# Recent Advances of Proteomics in Management of Acute Kidney Injury

**DOI:** 10.3390/diagnostics13162648

**Published:** 2023-08-11

**Authors:** Ilinka Pejchinovski, Sibel Turkkan, Martin Pejchinovski

**Affiliations:** 1Department of Quality Assurance, Nikkiso Europe GmbH, 30885 Langenhagen, Germany; ilieska.ilinka@hotmail.com (I.P.); sibelataolturkkan@gmail.com (S.T.); 2Department of Analytical Instruments Group, Thermo Fisher Scientific, 82110 Germering, Germany

**Keywords:** acute kidney injury, proteomics, biomarkers, clinical application

## Abstract

Acute Kidney Injury (AKI) is currently recognized as a life-threatening disease, leading to an exponential increase in morbidity and mortality worldwide. At present, AKI is characterized by a significant increase in serum creatinine (SCr) levels, typically followed by a sudden drop in glomerulus filtration rate (GFR). Changes in urine output are usually associated with the renal inability to excrete urea and other nitrogenous waste products, causing extracellular volume and electrolyte imbalances. Several molecular mechanisms were proposed to be affiliated with AKI development and progression, ultimately involving renal epithelium tubular cell-cycle arrest, inflammation, mitochondrial dysfunction, the inability to recover and regenerate proximal tubules, and impaired endothelial function. Diagnosis and prognosis using state-of-the-art clinical markers are often late and provide poor outcomes at disease onset. Inappropriate clinical assessment is a strong disease contributor, actively driving progression towards end stage renal disease (ESRD). Proteins, as the main functional and structural unit of the cell, provide the opportunity to monitor the disease on a molecular level. Changes in the proteomic profiles are pivotal for the expression of molecular pathways and disease pathogenesis. Introduction of highly-sensitive and innovative technology enabled the discovery of novel biomarkers for improved risk stratification, better and more cost-effective medical care for the ill patients and advanced personalized medicine. In line with those strategies, this review provides and discusses the latest findings of proteomic-based biomarkers and their prospective clinical application for AKI management.

## 1. Introduction

In the recent years, there has been a steady and substantial increase of patients suffering from acute kidney injury (AKI), affecting 13.3 million people worldwide with a mortality rate of up to 1.7 million deaths [1,2]. This complex disorder is defined by many pathophysiological distinct conditions, and it is still considered as under-recognized outcome, usually associated with secondary aetilogies like cardiovascular complications or sepsis [3]. By definition, AKI is characterized by a significant reduction of the renal function and a subsequent increase in serum creatinine levels (SCr  ≥  26.4 μmol/L), associated with short- and/or long-term complications. Usually, the early signs originate in the proximal tubular cells of the renal cortex, where symptoms are asymptomatic until disease progression is advanced [4]. The spectrum of kidney injuries is manifested within hours or a few days without reduced urine output. The outcome is extremely serious, causing the accumulation of unfiltrated waste blood products, impaired electrolyte homeostasis, and inflammation, which in turn, induce an imbalance of normal kidney function [5].

AKI is classified into three stages: prerenal, intrinsic renal, and/or postrenal. Prerenal renal injury is characterized by diminished renal blood flow, often due to hypovolemia, which leads to a decrease in glomerular filtration rate (60 to 70 percent of cases). In intrinsic renal injury, there is damage to the renal parenchyma, often from prolonged or severe renal hypoperfusion (25 to 40 percent of cases). The medical intervention, drug induced acute interstitial nephritis, accelerated hypertension, surgery correlated embolism, intrarenal deposition are considered as an intrinsic acute renal injuries. Postrenal injury occurs because of urinary tract obstruction due to tumor, benign prostatic hyperthropy or neurogenic bladder with decreased function of the urinary collection system (5 to 10 percent of cases) [5,6].

Nowadays, AKI management is of high importance due to the fact that clinical data are constantly showing an association with progressive loss of the kidney function and an increased risk of initiation of renal replacement therapy (RRT). The awareness of such a situation is evident because early recognition of AKI to improve kidney function and reduce long-term burdens is really at a moderate level. Lack of consistency and standardization in diagnostic classification for AKI has been an issue for real estimation of disease severity [7]. Current diagnosis based on patient history, physical examination, laboratory analysis, ultrasound, and kidney biopsy is limited due to non-early AKI detection and inability to predict disease course [7,8]. Often, this is associated with over-or-under treatment of the patients with dramatic increase in medical costs as well as a multifactorial unpleasant experience of physiological issues [9]. In addition, there is no approved medical therapy to prevent, treat, or enhance AKI recovery, which is a significant problem for the critically ill patients.

Within the last two decades, the study of proteomics has progressed enormously and most importantly, has revolutionized our understanding of molecular biology. Proteins and their smaller molecular units, called peptides, display the physiological and pathophysiological processes inside the cell or organism. This empowers us to utilize the complete set of proteins (proteome) to examine their structure, function, and expression in the cell, ultimately improving human health [10]. Proteome in general is highly dynamic and occasionally responds to different environmental stimuli. As we know, disease mechanisms and drug effects have a tremendous impact on the protein profiles, which is why it is important to reveal crucial information for an in-depth understanding of the disease and therapy on a molecular level [10,11].

Latest developments in high-resolution technologies enable high-speed levels and exceptional analytical performance designed for the assessment of complex biological samples. This in turn, has opened new avenues for the identification and characterization of novel biomarkers, especially in the field of proteomics and body fluids [12,13,14]. Proteins can be indicative of molecular changes during the disease state at first, and at the same time, might be a signal for disease progression. In fact, application of those molecular targets, features, and signatures in biomarker-guided therapies has been a major interest for the scientific community not only in the past few years but also it is the future prospective [15,16]. Especially, assessment of novel biomarkers for improved diagnostics but also prognostic accuracy, patient risk stratification, prediction of disease outcome, and monitoring of response to treatment are of special interest [17,18,19]. Therefore, a better and more comprehensive understanding of the protein’s dynamically driven biological functions, including metabolic cross-talk interactions, is an unmet need for a more precise understanding of disease onset and progression.

The aim of the review is to provide a comprehensive overview of the protein-based biomarkers that have been reported in the last 5 years, involved in diagnosis, prognosis, and monitoring treatment response at disease onset. 

## 2. Materials and Methods

### Study Design and Screening Procedure

Systematic literature screening for the last 5 years was performed using the PubMed database. The literature search was done in November 2022. Manuscripts was retrieved based on the following search criteria: “Proteomics” AND “Acute Kidney Injury” AND “Biomarkers”. The retrieved set of manuscripts were additionally prescreened based on the following inclusion criteria: (a) only original studies; and (b) human specimens. As exclusion criteria were set: (a) animal studies, (b) review manuscripts and (c) methodological manuscripts. The rest of the manuscripts were thoroughly investigated on type of the proteins reported as well as their association with disease progression. 

## 3. Results

The systematic review from 2018 to 2022 yielded 75 manuscripts that followed the screening criteria. The study design and workflow are presented in Figure 1. After the initial search of the titles, the manuscripts were further assessed by manual screening of the abstracts. This resulted in the selection of 12 original manuscripts that were included in this review. In addition, for a more clear and precise presentation of the results, the biomarkers are listed in Table 1 based on AKI categories and original manuscripts are briefly presented in Table 2 based on the most significant ones of the listed biomarkers. The other manuscripts that did not fulfill the above stated criteria were excluded. 

### 3.1. Acute Kidney Injury (AKI)—Related Protein Biomarkers

In this section, we present a short overview of the biomarkers that have been reported in the literature for the last 5 years, describing their involvement in molecular pathways and processes during disease onset.

In light of the three stages of AKI, the examined biomarkers are sorted into categories as prerenal, intrinsic renal injuries -intrinsic renal after medical intervention- and postrenal injuries. The biomarkers are also defined under three types as diagnostic, prognostic and monitoring biomarkers according to their characteristics, stated in recent studies. The proteins that are utilized to detect and confirm AKI are named as diagnostic AKI biomarkers. The ones that provide information on AKI stage and affected cells or areas of the kidney are named as prognostic, and the ones that support the research if the treatment effect is different for biomarker positive patients are classified as monitoring biomarkers. Together with the biomarkers, the affected kidney areas and cells are summarized in Table 1 based on the findings of the investigators. 

The results reveal that that focus of recent biomarker research has been on acute kidney injury development, particularly after treatments, e.g cancer therapy, cardiovascular surgery etc., to examine the effect of modern therapies (Figure 2).

### 3.2. AKI—Related Protein Biomarker Types and Their Association

A deep profiling of proteins involved in AKI development provides crucial evidence for the identification of new biomarkers and their classification as diagnostic, prognostic, or monitoring biomarkers. A special interest, in context to AKI, was protein-based biomarkers detectable in various bodyfluids, which likely can play a significant role in providing more specific, accurate, and medical knowledge for early diagnosis and future optimization of disease treatments. Efforts towards reaching these goals and all aspects of better patient management have been discussed under biomarker type subsections. The details of clinical studies, including their most important discoveries, are presented in Table 2.

### 3.3. Diagnostic AKI Biomarkers

β2 microglobulin (B2M) is a blood protein that is present on the surface of nucleated cells as a part of the normal immune system. This protein has a molecular weight of 12 kDa and is released by the cells into the blood, generally being a highly concentrated circulating protein compared to its lower levels or traces found in urine, the spinal cord, and other biofluids [32]. Until now, it is known that B2M associates and forms complexes together with the major histocompatibility complex I (MHC-I) and human leukocyte antigen I (HLA-I) on the cell surface [33]. Another important linkage is with the Fc receptor, which supports the regulation of immunoglobulins G, albumin, and hepcidin [34]. Previous experimental data suggested the potential implication of B2M in several diseases, not only in glomerulonephritis and/or AKI but also in hemochromatosis [35]. In the kidneys, *B2M* is usually filtered in the glomeruli, and then 99% of the content is reabsorbed in the renal proximal tubule structures. Higher concentration in urine could be detected, and this is due to renal impairment and the inability of proper protein reabsorption which leads to reduced renal function [33,36]. In other words, B2M is known as potential candidate for assessment of renal function in AKI and CKD and plays as mediator for some complications in uremic syndrome [33,35,36]. In the same time, this protein serve as a functional biomarker for studying and assessing the acute tubular necrosis (ATN) and fibrosis [37]. However, diagnosis and prognosis of AKI using *B2M* protein has poor outcome although some studies reported that applying machine learning modeling based on several biomarkers could improve the performance rate [16,20,32,35]. Using this approach, B2M has been investigated and was upregulated in CI-AKI patients but in relatively small patient cohort, meaning that more thorough examination and further validation in larger cohorts are needed [21].

Alpha-1-Antitrypsin (AAT) is one of the most abundant and active proteins, with a molecular size of 54 kDa. AAT belongs to the serine protease inhibitor family (also known under the SERPIN acronym), which is mainly produced in the liver. It was initially discovered in human plasma as a glycoprotein that was characterized to have inhibitory effect on several proteases, including elastase and/or proteinase-3 [38,39]. Due to its inhibitory effect, AAT can cause anti-inflammatory effects and improvement of injured tissue during evaluated molecular pathways. Despite the molecular function, AAT has a high affinity for building complexes with hemin salt in order to prevent forming of porphryias and hemin-induced reactive species in neutrophils [40,41]. Regarding AKI, it has been previously reported that AAT was identified in urine and considered as a biomarker for ischemic injury [16,42]. Several in vivo and in vitro reports defined the AAT’s cellular targets during the adaptive immune response, mainly being macrophages, B-lymphocytes, and dendritic cells [43,44]. However, the complete mechanism of AAT action is not entirely discovered, although some studies have shown interesting data on ATT profound role during a series of nephrotoxic and ischemic injuries [45,46,47].

Retinol-binding protein (RBP) is a low-molecular-weight protein with a molecular mass of 21 kDa [48]. It is known as a circulating plasma protein synthesized in the liver which is responsible for the transportation of the fat-soluble vitamin A, called retinol. RBP forms complexes with transthyretin and preserves glomerular filtration in the kidneys [48,49]. The majority, or approximately 95%, of the protein is reabsorbed in the proximal tubule. Only a small portion (4–5%) of serum RBP freely passes the glomerular barrier and is excreted in urine [50,51]. Urinary RBP is relatively stable in urine and therefore serves as a diagnostic marker for tubular dysfunction and tubularpathies [52,53]. Lately, urinary RBP has been studied as a biomarker for AKI patients suffering from heart diseases [21].

Fibrinogen (FBG) is another large polypeptide belonging to the group of glycans (glycoproteins) found in plasma. It has a molecular weight of 340 kDa and is synthetized in the liver. Structurally, fibrinogen is composed of three different polypeptide chains, namely alpha, beta, and gamma chains, which are linked through disulfide bridges to develop the stable chemical structure of the fibrinogen molecules [54,55]. In return, fibrinogen creates structural complexes with other molecules and represents one of the major proteins responsible for blood clotting. Moreover, FBG plays a crucial role in the regulation and activation of coagulation proteins during platelet aggregation. This process is largely supported by fibrin’s products, Fibrinopeptide A and B proteins, and activated factor XIII (FXIII), which are forming solid clot network [56,57]. However, fibrinogen has been associated with some serious medical condition, not only systematic inflammation but also with AKI [58]. Increase accumulation of fibrinogen was related with AKI development after cardiac or abdominal aortic aneurysm repair surgery [59]. In addition, detection and identification of fibrinogen could be risk sign for renal inflammation and damage, such as infection, early graft dysfunction, graft failure or even death [60,61,62]. Based on the latest case-control studies, fibrinogen has been particularly implicated as a sensitive biomarker for early AKI diagnosis and was suggested to have beneficial role in estimating the risk for AKI after cardiac or liver transplantation surgery [23,59,60]. 

### 3.4. Prognostic AKI Biomarkers

*Cystatin C (CysC)* is a small and low-molecular-weight protein with a molecular mass of 13 kDa. It consists of 122 amino acids, and because of its small size, protein levels can be freely reabsorbed by the glomerulus and metabolized after reabsorption [63]. Cystatin C is a protein that is produced mainly by nucleated cells at a constant rate. It’s known as a member of the cysteine protease inhibitors released in the blood system [64]. In general, CysC concentration can be detected in urine and plasma. Urinary CysC is a biomarker for the deterioration of proximal tubular cells, and its potential has been studied in terms of the prediction of kidney injury and its prognosis. Experimental investigation of the urinary CysC’s role has confirmed its capability to provide early signs of renal impairment compared to well-known clinical markers like creatinine [65]. In contrast, serum CysC has been monitored in several clinical studies, showing a strong association with GFR estimation [66,67]. In fact, higher levels of serum CysC and a decline of the GFR function are closely linked, hence representing a reference standard for kidney injury in critically ill AKI patients [64,68]. In addition, increased concentration of serum CysC has been also associated with increased HbA1c levels in diabetic patients who did not show any signs of kidney damage [69]. Overall, future studies incorporating CysC in clinical practice for improved patient management would be highly beneficial.

S100 calcium-binding protein P (S100P) is a member of the S100 family of proteins and contains helix-loop-helix (EF-hand) Ca^2+^-binding motifs. It has a molecular mass of 10 kDa, and it is expressed in various organs like the human placenta, stomach, urinary bladder, and bone marrow [70,71]. As a result of its molecular structure, S100P actively regulates calcium homeostasis and, at the same time, acts as a calcium signaling molecule. In addition to its main function, S100P has the ability to communicate and bind with other proteins involved in the regulation of actin cytoskeleton dynamics and extracellular matrix (ECM) synthesis and degradation [72]. Experimental data on S100P suggested a possible interaction with other proteins like Erzin, myosin IIA, cathepsin D, and Ras GTPase-activating-like proteins [73,74,75,76]. S100P has also been reported as a biomarker for various cancer types [77,78,79]. In regard to AKI, S100P proteins levels were investigated in a relatively small cohort of 24 patients suffering from contrast-induced acute injury (CI-AKI), and therefore, significant up-regulation of the S100P in CI-AKI patients compared to the non-CI-AKI group [21]. Similarly, significant increased levels were found in the prediction of AKI in preterm infants [24]. However, further assessment of the S100P role in prediction of AKI are warranted.

Galectin-3-binding protein (Gal-3BP), also known as 90k or Mac-2, is a glycoprotein secreted in the body fluids. As a member of the glycoprotein family, Gal-3BP is a typical protein prone to modifications with other amino acids and was initially identified to be involved in cellular transformation in different cancers [80,81,82,83,84,85,86,87]. It contains a carbohydrate recognition domain (CRD), which allows molecules and enzymes to oligomerize and form pentamers macromolecules. The molecular mass of Gal-3BP is calculated 65.3 kDa, although some reports indicate that the secreted form of the protein into bodyfluids could reach up to 100 kDa [88]. Functionally, Gal-3BP recognizes and acts as a receptor for small pathogen-associated molecular patterns (PAMPs) and damage-associated molecular patterns (PAMPs) in their transportation through the cytosol and external microenvironment of the cell, including gene regulation [89]. However, his role has also been investigated in terms of inflammation and the process of fibrosis [90]. Most interesting, this glycoprotein is well-known for its involvement in the synthesis of proinflammatory cytokines and the homeostasis of reactive oxygen species in AKI [91]. To elucidate more on the cause of the kidney injury as well as to gain novel insights on the biomarkers sensitive on drug-induced kidney injury, urinary Gal-3BP has been found to be upregulated in nephrotoxic injury due to vancomycin as the prototypical renal toxicant, remaining a biomarker of future interest for clinical assessment [23].

Alpha-2-microglobulin (α_2_M) is a tetrameric protein that has a molecular size of 720 kDa and is one of the most abundant proteins found in blood plasma. *α*_2_M has a special ability to inhibit different kinds of proteases, affecting distinct biological processes not only in plasma but also in cerebral spinal fluid, spinal fluid, synovial fluid, ocular fluid, and interstitial fluid [92,93,94,95]. The mechanism of action relies on the formation of tetrameric ’traps’, which disturb the function of the active proteases with the actual substrates [96]. In this way, ‘trapped’ proteases are exempted from molecular digestion of collagens and other large proteins/peptides, preventing conformational changes of small polypeptides entering into the ‘trap’. This so called ‘selectivity’ of *α*_2_M protein is based on the presence of a polypeptide ‘target’ region in the chemical structure of *α*_2_M which became attractive for most of the proteolytic peptidases [96,97,98,99]. In the past, most of the studies related to *α*_2_M role were focused in exploring hemostasis and thrombosis as critical conditions for patients in intensive care units (ICUs) [100,101]. However, upregulated *α*_2_M levels have also been associated with other serious chronic diseases: kidney-related and cardiovascular related diseases [102,103,104,105,106,107,108,109,110]. Clinically, *α*_2_M has been used as a biomarker for monitoring treatment response as well as to elucidate the pathophysiological mechanism of kidney injury in AKI [23].

CD26 protein, also known as dipeptidyl peptidase-IV (DPP4), belongs to the group of glycoproteins that are expressed in epithelial cells in the liver, kidney, and intestine. Based on some previous laboratory measurements, CD26 is a large protein with a molecular size of 110 kDa [111]. As an enzyme, it has the ability to hydrolyze amino acids like proline and/or alanine from N-terminal residue. At the same time, CD26 can regulate and enhance the expression of T-cells during signal transduction pathway activity [112]. In addition, there is evidence of its involvement in immune memory cells, specifically in regulating and balancing CD4 T-cells and providing an appropriate immune response when immunological treat occur [113,114]. In regard to the clinical utility of this biomarker, CD26 has been investigated towards ischemia-reperfusion injury in the kidneys and its potential role in suppression of apoptosis and inflammation. Moreover, CD26/DPP-4 inhibitors have been assessed and investigated for their nephroprotective role in glycemic management to reduce the incidence of micro- or macroalbuminuria in type 2 diabetes patients [115,116,117]. Lately, urinary exosomal CD26 has been investigated to see whether it can predict renal reversal and recovery from AKI. In a single-center cohort of 200 patients, CD26 was negatively associated with AKI, including major adverse or worsening of renal function, but its expression was positively associated with renal improvement and recovery. In addition, the prediction value of CD26 was confirmed for early reversal and recovery in non-septic AKI, hence demonstrating the prediction potential of this biomarker for clinical studies [25].

Complement C3 (C3) is a large plasma protein with a molecular weight of 180 kDa [118]. As one of the most abundant members of the complement system, C3 has a central role in the innate immune system, in which detection and elimination of foreign molecules by activation of macrophages occur. This process is carried out by complement cascade mobilization when C3 associates with and forms complexes with other amino acids in the host [119,120]. Usually, C3 is activated by several pathways (classical, alternative, or lectin), which ultimately trigger the defensive removal action for pathogens, debris, or cellular structure [121]. The protein activation also stimulates C3 cleavage into two fragments (C3a and C3b), which later remain attached to the cell surface and act as signaling molecules in cell homeostasis [122]. Interestingly, C3 has been reported as a protein involved in a variety of kidney diseases, including AKI [123,124,125,126,127]. Evaluated serum C3 levels have been associated with reperfusion injury in AKI through cascade activation and the process of autophagy [128,129]. These ongoing molecular processes might influence various functional pathways, most specifically energy assumption, inflammation, cell-division cycle, and lipid modifications. As a result of such changes, there is a possibility that some C3 protein fragments may be secreted or cleaved into urine and serve as potential diagnostic or prognostic biomarkers [129,130,131]. Recently, a novel approach utilizing urinary exosomal proteomics has been performed to characterize the pathophysiological mechanisms involved in drug-induced kidney injury. Based on the presented data, C3 and C4 were the proteins that showed the most significant performance in the stratification of drug-induced patients from healthy individuals [23].

### 3.5. Monitoring AKI Biomarkers

Tumor necrosis factor (TNF) is a cytokine with a molecular mass of 25.6 kDa. This molecule is secreted by the macrophages, and it is well-known for its central role during inflammation and stress response cascades. TNF-α has a high affinity for binding with other receptors, and therefore it can be found in a soluble and membrane-bounded form. Usually, it is in close interaction with the soluble receptors (sTNFR1 and sTNFR2), and its main function is related to host defense, cell proliferation, and differentiation. In terms of AKI, experimental studies showed associations with sepsis, sepsis shock, and the inflammatory response during severe conditions as well as in autoimmune diseases [132]. Functional analysis indicated TNF-α involvement in hypertension and mediation of blood pressure and NaCl retention [133]. Lately, sTNFR1 and sTNFR2 as circulating plasma biomarkers were assessed to investigate the potential to stratify patients who are at low or high risk for CKD progression after 3 years of hospitalization during AKI onset. Associations between biomarkers and kidney disease progression were evaluated in 500 patients using multivariable logistic regression models [22]. Although sTNFR1 and sTNFR2 have been observed to be related to AKI pathophysiology, they are not specific AKI biomarkers, and therefore more validation studies are needed to prove their clinical utility.

Annexins represent another group of large proteins with a molecular mass of between 33–39 kDa. They are characterized as a protein superfamily that shares a very similar homological structure among all members. As a result of their core structure, they have an affinity to bind phospholipid- and calcium-based proteins [134,135]. There are more than 1000 proteins identified in different species, but in humans only 12. Moreover, annexins have been recognized as intracellular and extracellular proteins that can be attached to various cellular membrane ligands and receptors, involved in coagulation, inflammation, and molecular transport through membranes [136]. Evaluation of the annexin levels have been also assessed in the prediction of AKI progression, particularly when ischemic renal dysfunction significantly contributes towards apoptosis. Importantly, the latest proteomic investigation performed in patients with CI-AKI showed significant levels of annexins A1, A2, and A3 [21]. Annexin A1, also known as lipocortin, has been mainly controlled by glucocorticoids hormons and cytosolic phospholipase A2 (PLA2) inhibition, responsible for the mechanism of arachidonic acid (ARA) release and the prevention of signaling molecules (eicosanoids) synthesis [137,138,139]. The protein itself has binding affinity to the phospholipid layers of the cell membranes, hence making it a molecular target for monitoring treatment response through cross-linkage with formyl peptide receptors [140]. With similar properties, cytoplasmatic annexin A2 has been largerly studied for its roles in intracellular regulation and cell signaling cascades [141]. Latest reports have demonstrated that there is a link of annexin A2 involvement during glomerulosclerosis by which collagen type VI is secreted via renal glomerular endothelial cells [142]. Interestingly, in another study, increased annexin A2 levels were observed in injured proximal tubular epithelial cells mediated by calcium oxalate salt, thus becoming a potential biomarker for monitoring tubulointerstitial inflammation of the kidneys [21,142,143].

Ostepontin (OPN) is another type of glycoprotein with a molecular mass of 35 kDa [144]. OPN is prone to post-translational modifications (PTMs), especially phosphorylation on serine and threonine, O-glycosylation, and transglutamination, which may enlarge the molecular size up to 75 kDa [145]. It is also known as a pleotropic glycoprotein expressed in a variety of cells throughout the human body, such as activated T cells, macrophages, natural killer (NK) cells, neutrophils, and dendritic cells, among many others [146]. OPN has a crucial role in the formation of the bone and during bone metabolism. Normally, it is upregulated during inflammation and plays the role of a regulator of the immune response. During normal conditions, OPN is present in the loop of Henle and in the nephrons of the kidneys. After renal damage, significant expression of OPN may be observed in the urine as a result of tubular and glomerular deterioration [147,148,149]. In this line, it is important to mention that OPN function is also related to inhibition of bone mineralization, more specifically in prevention of kidney stone formation [148]. From a clinical point of view, the latest proteomics-based approaches in addition to machine learning algorithms could predict AKI risk in patients undergoing coronary angiographic procedures in a cohort of 889 patients with various acute and non-acute indications, from which 4.8% developed AKI [20]. In relatively smaller patient’s cohorts, serum OPN showed its potential for predicting the outcome of renal replacement therapy as well as the mortality rate in very-low-weight-birth infants [150,151].

C-reactive protein (CRP) is a blood-circulating protein with a molecular mass of 115 kDa that is a member of the pentraxin protein family [152]. CRP has been mainly known as a protein that could be sensitive to a variety of changes in the host due to inflammation, infection, or trauma, including additional pathophysiological changes during tissue damage [153]. In the diagnostic area, it is recognized and used as a clinical marker for inflammation during cardiovascular and kidney diseases [154,155]. Based on the previous laboratory data, CRP is primarily synthetized in the liver and also in smooth muscle cells, macrophages, endothelial cells, and some other types of cells [156]. Apparently, the major role is activating the C1q molecule in the complement pathway, interacting with cell-mediated pathways in the complement pathway, and binding to Fc receptors of IgG in order to release pro-inflammatory cytokines [157,158]. In regard to the AKI, it has been reported that CRP concentrations were evaluated and were significantly correlated with serum creatinine and urea concentrations during AKI onset [159]. In fact, the evidence suggests that after AKI improvement, there has been a significant drop in CRP’s concertation. On the other hand, CRP has been studied and confirmed its role as an independent predictor of higher AKI mortality in older patients [160]. Recently, a proteomic-supported biomarker classifier including CRP has been evaluated in order to test its prediction accuracy for patients undergoing coronary angiography [20]. CRP’s role has been investigated in AKI patients undergoing surgery procedures to treat coronary artery bypass grafting (CABG) and the ability for pre- and post-operative patient stratification [161].

Complement C4 (C4) is another essential protein from the complement system. It has a molecular mass of 206 kDa. C4 plays a crucial role in the activation of classical and lectin complement cascades, generating a defense system that is constantly attacking and eliminating different kinds of pathogen organisms [162,163]. As a result of this function, tissue homeostasis is preserved. From a chemical point of view, C4 is a glycoprotein, and it is built up from three subunits: α, β and γ -chain, which are linked together via disulfide bonds [164,165]. Generally, C4 activation is related to the activation of other proteins from the complement system, inducing an increase in the synthesis of different signaling molecules, most specifically cytokines and chemokines. In this way, significant recognition of the unknown organisms is performed, hence providing a long-lasting cell-mediated and humoral immune response [166,167,168]. Moreover, excessive investigation of the complement proteins, including C4, has revealed the potential for monitoring renal disease pathogenesis and patient stratification [169,170,171].

Transferrins (TF) are a group of glycoproteins that are responsible for the transportation of Fe (III) from the cardiovascular system to multiple cells and tissues inside the human body. Until now, three types of transferrins (serumferrin, ovoferrin, and lactoferrin) and one type of trans-transferrin (ferric ion-binding protein) have been characterized [172,173]. Serotransferrin is known as the fourth most abundant nonheme iron-transport glycoprotein, build up of 679 amino acid residues that maintain iron homeostasis. It has a molecular weight of 76 kDa and is synthesized in the liver [174,175]. This protein is fully glycosylated, hence having two N-linked glycosylation sites: Asn^432^ (Asn-Lys-Ser) and Asn^630^ (Asn-Val-Thr), respectively. Since glycosylation is site-specific, any changes in the structure of the serotransferrin during disease states could cause the formation of fucosylated glycans [176,177]. In addition, deficiencies of the protein lead to a shortage in the transportation of iron in the cells, subsequently suppressing cellular proliferation [178]. In clinical practice, the prognostic value of serotransferrin and other prognostic circulating iron markers have been assessed. Based on the reported data, it was found association with mortality in critically ill patients with severe AKI requiring renal replacement therapy (RRT) [179]. In line with clinical manifestation, increased urinary levels of serotransferrin might be indicative of a risk of renal dysfunction and further used as a biomarker for patent stratifications with a risk of nephrotoxic injury after drug treatment [23,180,181].

Complement factor B (CFB) is known as a protein that has glycan structures covalently attached to the main polypeptide chain. It is mainly produced in the liver and is present in the plasma with a molecular mass of 93 kDa [182]. CFB is a required protein and is involved in the activation and amplification of the alternative pathway (AP) of the complement system. The process starts when CFB binds to C3b, which is located on the cell membrane, and forms a molecular complex called C3b,B. This complex is then disturbed by the action of CFD, generating a Bb residue bond to C3b. Later, C3/C5 convertase does the proteolytic cleavage and produces components like C3a, C5a, C3b, and C5b, respectively. Additionally, CFB is involved in the process called “tickover”- spontaneous hydrolysis of thioester that form C3(H_2_O) in the amplification loop [182,183]. Several experimental studies showed that AKI is associated with an inflammatory response that leads to kidney injury, systematic inflammation, and ischemia/reperfusion injury (IRI). All of this can contribute to significantly worse renal function and even the development of cardiovascular diseases [184,185,186]. An alternative pathway of the complement is becoming triggered in the presence of cardiopulmonary bypass (CPB). This is when blood comes in interaction with artificial materials, potentially prolonging ischemic AKI and severe renal injury [187]. Therefore, urinary CFB as a pre-operative biomarker has been identified and further assessed in combination with the current clinical scoring to improve prediction of AKI after cardiac surgery [26]. Proteomic analysis in a relatively small cohort of AKI patients enhanced the current post-surgery AKI risk stratification, but larger studies are definitely needed in order to confirm the ability of CFB in the prediction of severe renal outcomes.

Histidine rich glycoprotein (HRG) is a plasma glycoprotein present in relatively high concentrations. The molecular mass is in the range of 75 kDa and is synthesized by the liver parenchymal cells as well as by monocytes and macrophages [26]. HRG has a multi-domain polypeptide structure, which allows to perform dynamic interconnection with a variety of ligands (heme, divalent cations, heparin, heparin sulfate, tropomyosin, plasminogen, fibrinogen, and some others) and members of the complement system (C1q, factor H, C8, C4, C3). Some studies suggest that HRG is a multifunction protein that can regulate the formation and clearance of immune complexes, dead cells, and pathogens [188,189]. Additionally, HRG is involved in the molecular processes related to cell adhesion, angiogenesis, coagulation, and tumor progression [190]. However, HRG association with AKI is currently unknown and requires further investigation. For that reason, a study using high resolution mass spectrometry proteomic data found an association of HRG with postcardiac surgery in AKI. This study was conducted in a small patient cohort. Interestingly, the prognostic biomarker discovery was validated by using ELISA, by which combination of clinical scoring and biomarkers for AKI prediction showed improvement in patient stratification, aiming at better AKI management and treatment in clinical settings [190].

Serine proteinase kallikrein (KRK1) is a secretory product synthesized by the exocrine glands (pancreas and salivary glands). Usually, this enzyme is expressed in a wide range of tissues across the human body and is actively involved in various kinds of clinical complications [191]. When activated, KRK 1 has an estimated molecular mass of 29 kDa and it is a member of the largest cluster of serine proteases family. The main molecular function of KRK 1, is to perform proteolytic digestion on the low molecular weight kininogen (Arg–Ser and Met–Lys bonds) to generate another type of component called kallidin (also known as lysyl-bradykinin). In this form, kallidin is bound to specific receptors and exhibits diverse regulation of different pathological complications. Despite the primary function of KRK 1, this enzyme can also cleave kallistatin, somatostatin, pro-insulin, low-density lipoprotein, and some other proteins [192,193,194]. Before being secreted into urine, KRK 1 remains located inside renal tubules and suppresses the activity of sodium channels, H^+^, K^+^-ATPase. Despite of deeper knowledge of KRK 1, the protease is still subject of extensive investigation regarding its clinical utility [27,195]. For that reason, increased activity of KRK 1 was identified when urine proteomes from AKI patients at 1-hour CBP and at ICU (serum creatinine >50% from baseline or 26.5 μmol/L) were compared to non-AKI patients [26]. Collectively, urinary kallikrein activity was detected in both patients’ groups, suggesting that the role of kallikrein might be related to renoprotective effect in human AKI, but this hypothesis needs further validation in a larger patient cohort.

## 4. Conclusions

Proteomic-based studies summarized in this review (Table 2) emphasized the importance of proteins and their unique structure and function in the context of AKI, aiming at maximal utilization of different analytical platforms for biomarker identifications and gathering diverse molecular information for improvement of disease pathophysiology. The studies further elaborated on crucial knowledge of critical proteins and their alternation in AKI in order to identify reliable biomarkers to guide and enhance patient management. Another key point, based on the data presented, is the deployment of machine learning algorithms for advanced investigation of the diagnostic/prognostic values of the protein profile, as well as the possibility of their application for monitoring of drug treatment.

During the last 5 years, the scope has been on investigation of AKI-related biomarkers and their association with disease severity (Table 1 and Table 2). Relatively novel AKI biomarkers like sTNFR1, sTNFR2, S100P, CD 26 and KRK1 in urine and plasma have been introduced, hence showing some promising results in AKI stratification and their association with severity and progression. Studies with these biomarkers provided some novel molecular pathway information and interactions for unrevealing mechanisms of kidney injury events. Despite these investigations, the clinical application and relevance of these and other biomarkers in AKI is still inclusive [28,29].

## 5. Summary and Outlook

The path of broader biomarker application in clinical settings is still not ideal and straightforward. This can be also seen throughout this review, where majority of the studies were conducted in relatively small patients’ cohorts with few exceptions [30,31]. However, the reviewed studies also revealed that without proper multi-centric validation cohorts it is almost impossible to have implementation of proteomic biomarkers for clinical applications. Characterization of protein expression and protein modification is challenging and requires high-throughput technology for biomarker discovery, validation, and drug development [196]. Regardless of their application, mass spectrometry (MS) has developed continuously, hence providing the opportunity for faster throughput for top-down and bottom–up MS-based methods, deeper proteome coverage and higher sensitivity in measurements of thousands of proteins in thousands of samples. This is achieved by combining different mass analyzers, Quadrupole Oribitrap with the asymmetric track lossless (Astral) analyzer, providing >200 Hz MS/MS scanning speed, high sensitivity, and a narrow mass accuracy window [197]. Protein pathway array (PPA) is another proteomic platform used for screening and analysis of protein signatures that control cancer development. This is a gel-based technique that is mainly used for tissue proteomics, where antibody mixtures are employed to detect specific antigens in protein samples derived from a biopsy or tissue [198]. In addition, next generation tissue microarray (ngTMA) technology was recently introduced. Despite the traditional tissue microarrays (TMAs), which are based on large-scale antibody examination using many small formalin-fixed paraffin-embedded and/or frozen blocks under identical conditions, ngTMA allows automatic tissue microarray analysis by carrying out digital pathology for improved clinical research and reducing the time needed for analysis [199]. Translation of the biomarkers from medical research to clinical use has been improved by the use of multiplex bead and/or aptamer-based assays. The whole concept depends on flexible fluorescent-labeled beads, specially coated with different antibodies, to identify an antigen or certain mutations in a sample [200]. However, there are also multiplex, single plex or ultrasensitive bead-based arrays such as Meso-scale Discovery (MSD) for cytokine detection as well as Single Molecule Array (SMA) for biomarker detection in certain diseases [201,202]. Last but not least, proximity extension assay is promising proteomic tool based on oligonucleotide-linked antibody pairs that have similarity between each other and are brought into proximity [203].

Utilization of more innovative proteomics strategies enabled more precise resolution of issues related to unbiased, accurate quantification, data processing, and clinical translatability. Although the goal of such studies attempting to discover AKI biomarkers is to identify and characterize one universal urinary or serum biomarker for determination of risk, diagnosis and prognosis of AKI severity, the optimism about this approach is slowly “coming to end”. One example is Neutrophil gelatinase-associate lipocalin (NGAL) also known as siderocalin or lipocalin 2 (LCN 2) who is an iron-carrying protein with molecular mass of 25 kDa. NGAL can be detected in urine and plasma [204]. Several studies confirmed the NGAL ability of proactive action in immune response to bacteria, prevention of apoptosis and proliferation of renal tubular cells. In fact, this represents one of the possible molecular pathways for preservation and defense of the kidney normal homeostasis during AKI onset, but it is non-specific biomarker related to other renal and non-renal conditions [205,206].

To proof validity and utility, nowadays supervised and unsupervised machine learning (ML) algorithms are more applicable for biomarker discovery and validation due to fact that larger studies with heterogeneous population could not confirm the excellent achievements in initial, small-scale, pilot studies with homogenous population [207]. Simultaneously, the generation of high-resolution proteomic data has largely increased the computational power and clearly demonstrated the utility of ML algorithm deployment in various applications, especially in the identification of protein signatures as well as the prediction of disease course based on proteome and clinical data.

On the other hand, extracting meaningful biological information from the large dataset using ML is challenging. Most importantly, common pitfalls include imprecise study design with uncertain primary or secondary outcomes and data that are biased and not harmonized across different patient cohorts during biomarker discovery and validation studies. Another problem is data selection and data integration when multiple sources are available, having data that could contain redundant information and could be misleading in decision-making approaches. This is usually related to inappropriate handling of biomedical raw data influenced by pre-analytical errors, resulting in variation of measured signals [207,208]. Preprocessing and filtering methods for the clinical data could be one of the solutions for data normalization and the occurrence of missing values. For comparative evaluation, appropriate ML algorithms able to deal with the number and types of input and output features need to be considered. This, at the same time, increases the risk of generation classifiers that overfit high-dimensional data and provide statistical inaccuracies. Therefore, to avoid such an outcome, optimization of the classifier parameters and feature selection is definitely pre-requested [209]. Finally, as indicated in the review, the lack of sensitivity of the biomarker classifiers in an external validation cohort is a major hurdle for diagnostic applications.

Collectively, future biomarker research studies (Figure 3) are expected to address these ongoing challenges with the aim of improving medical care in general and for critically ill patients. 

## Figures and Tables

**Figure 1 diagnostics-13-02648-f001:**
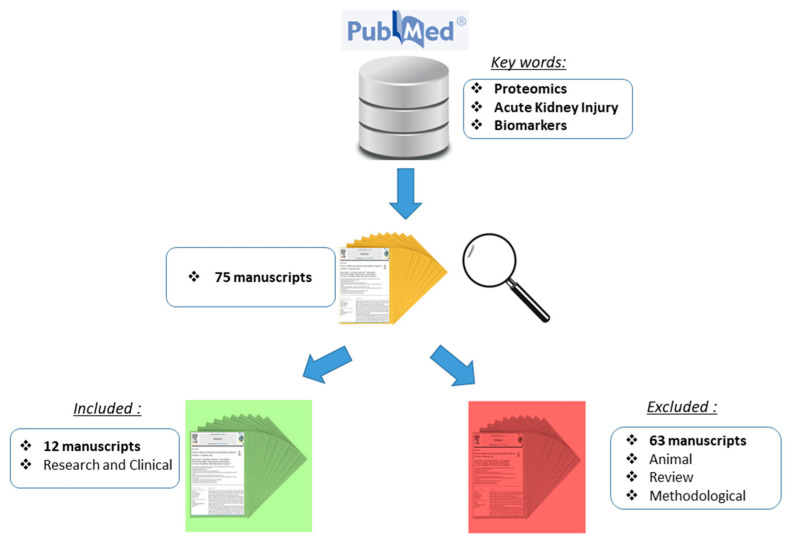
Schematic representation of the study design and literature search.

**Figure 2 diagnostics-13-02648-f002:**
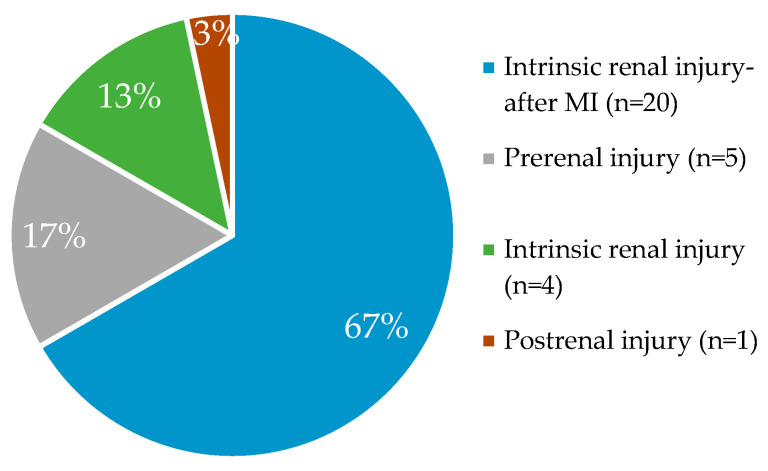
The percentage of biomarker distribution in examined studies based on AKI type.

**Figure 3 diagnostics-13-02648-f003:**
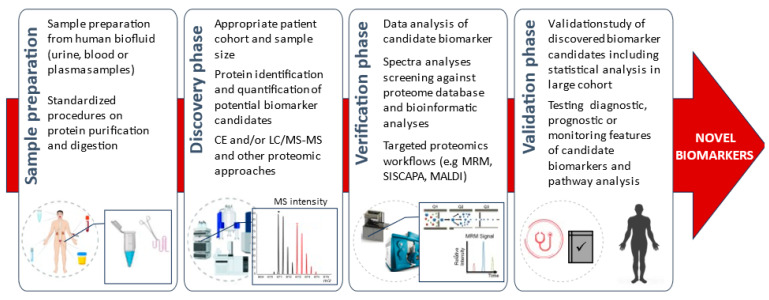
Workflow for potential biomarker discovery and validation.

**Table 1 diagnostics-13-02648-t001:** The list of biomarkers evaluated in AKI clinical studies.

Biomarkers	Biomarker Type	Study Type	Affected Area of Kidney	Affected Kidney Cell Types	AKI Category
NGAL	Diagnostic	Urine analysis	Renal pelvis	Collecting duct epithelial cells	Prerenal
B2M	Diagnostic	Urine analysis	Proximal tubule	Tubular epithelial cells	Intrinsic renal
SERPINA1 (AAT)	Diagnostic	Urine analysis	Proximal tubule	Tubular epithelial cells	Intrinsic renal
RBP4	Diagnostic	Plasma analysis	Proximal tubule	Tubular epithelial cells	Postrenal
FBG	Diagnostic	Urine analysis	Glomerulus	Tubular epithelial cells	Intrinsic renal (after MI) **
GDF15	Diagnostic	Urine analysis	Nephron	Renal endothelial cells	Intrinsic renal (after MI) **
LRG1	Diagnostic	Urine analysis	Nephron	Renal endothelial cells	Intrinsic renal (after MI) **
SPP1	Diagnostic	Urine analysis	Nephron	Renal endothelial cells	Intrinsic renal (after MI) **
ANXA5	Diagnostic	Urine analysis	Nephron	Renal endothelial cells	Prerenal
6-PGLS	Diagnostic	Urine analysis	Nephron	Renal endothelial cells	Prerenal
TIMP-2 IGFBP7 *	Diagnostic	Urine/serum	Proximal tubule	Proximal tubular epithelial cells	Intrinsic renal (after MI) **
C3	Diagnostic or prognostic	Urine analysis	Glomerulus	Tubular epithelial cells	Intrinsic renal (after MI) **
C4	Diagnostic or prognostic	Urine analysis	Glomerulus	Tubular epithelial cells	Intrinsic renal (after MI) **
GAL-3BP	Prognostic	Urine analysis	Glomerulus	Tubular epithelial cells	Intrinsic renal (after MI) **
Cys C	Prognostic	Plasma analysis	Proximal tubule	Tubular epithelial cells	Prerenal
S100P	Prognostic	Urine analysis	Glomerulus	Urothelium cells	Prerenal
α2M	Prognostic	Urine analysis	Glomerulus	Tubular epithelial cells	Intrinsic renal (after MI) **
CD26 *	Prognostic	Urine analysis	Glomerulus/Proximal tubule	Renal brush border epithelium	Intrinsic renal (after MI) **
sTNFR1, sTNFR2	Monitoring	Plasma analysis	Glomerulus	Tubular epithelial & mesangial cells	Intrinsic renal
ANXA-2	Monitoring	Urine analysis	Glomerulus	Renal glomerular endothelial cells	Intrinsic renal
CRP	Monitoring	Blood analysis	Renal cortex	Renal Cortical Epithelial Cells	Intrinsic renal (after MI) **
OPN	Monitoring	Blood analysis	Nephron-loop of Henle	Renal epithelial cells	Intrinsic renal (after MI) **
CD5 & Factor VII *	Monitoring	Blood analysis	Nephron	Filtrating cells	Intrinsic renal (after MI) **
IgHM	Monitoring	Urine analysis	Glomerulus	Tubular epithelial cells	Intrinsic renal (after MI) **
Serotransferrin	Monitoring	Urine analysis	Glomerulus	Tubular epithelial cells	Intrinsic renal (after MI) **
HRG	Monitoring	Urine analysis	Glomerulus	Proximal tubule epithelial cells	Intrinsic renal (after MI) **
CFB	Monitoring	Urine analysis	Glomerulus	Proximal tubule epithelial cells	Intrinsic renal (after MI) **
CD59	Monitoring	Urine analysis	Glomerulus/Proximal tubule	Renal brush border epithelium	Intrinsic renal (after MI) **
AGT	Monitoring	Urine analysis	Glomerulus	Proximal tubule epithelial cells	Intrinsic renal (after MI) **
KRK1 *	Monitoring	Urine analysis	Glomerulus	Proximal tubule epithelial cells	Intrinsic renal (after MI) **

* indicates downregulation of protein biomarkers associated with disease outcome ** denotes for intrinsic renal injury after medical intervention.

**Table 2 diagnostics-13-02648-t002:** Characteristics of the AKI clinical studies and their most important discoveries.

Authors	Biofluid	Method	Patient Cohort	Investigated Biomarkers	Most Significant Biomarkers	Conclusion
Ibrahim et al. [20,21]	Blood	Luminex xMAP immunoassay	44 AKI 745 non- AKI	109	CRP; OPN; CD5; FACTOR VII	The biomarker panel using machine learning was developed and showed a performance with an AUC of 0.79 for predicting procedural AKI. The optimal score cutoff had 77% sensitivity, 75% specificity, and a negative predictive value of 98% for procedural AKI. An elevated score was predictive of procedural AKI in all subjects (odds ratio = 9.87; *p* < 0.001).
Zhu et al. [21,22]	Urine	LC-MS/MS	4 CI-AKI 20 CI-non AKI	99	NGAL; S100- P; ANXA2; B2M; SERPINA1; RBP4	In relatively small patient cohort, urine proteome of CI-AKI vs. non-CI-AKI were compared. Upregulation was observed in CI-AKI with ratio of 7.40 (B2M), 6.63(S100-P), 4.25 (NGAL) and 4.27 (SERPINA1).
Awdishu et al. [23]	Urine/blood	LC-MS/MS	10 V-AKI 12 HC	251	C3; C4; GAL-3BP, FBG, α2M; IgHM; SEROTRANSFERRIN	Urinary exosome proteins in response to V-AKI might provide vulnerable molecular information that helps elucidate mechanisms of injury and identify novel biomarkers among patients with confirmed drug-induced kidney injury.
Jung et al. [24]	Urine	LC-MS/MS	14 AKI 14 non-AKI	174	NGAL; ANXA5;GAL3; 6-PGLS; S100-P	Proteomic urinary-based biomarkers that can predict early AKI occurrences in infants were identified. Three biomarkers performed well, showing AUC values of 0.75, 0.88 and 0.74 for NGAL, ANXA5 and S100-P, respectively. There was higher beneficial effect of the classifier performance when NGAL + AXA5 (AUC of 0.92) and NGAL + AXA5 + S100-P (AUC of 0.93) were applied.
Du et al. [25]	Urine	Flow cytometry	133 AKI 68 non-AKI	1	CD26	Urinary exosomal CD26 was negatively correlated with AKI compared with non-AKI patients (β = −15.95, *p* < 0.001). Similar results were obtained for the AKI cohort with major adverse events. On the other hand, AKI survivors exhibited high-CD26 levels compared AKI patients with low-CD26 levels for early reversal, recovery and reversal, respectively, after adjustment for clinical factors (ORs (95% CI) were 4.73 (1.77–11.48), 5.23 (1.72–13.95) and 6.73 (2.00–19.67), respectively). Prediction performance was moderate for AKI survivors (AUC 0.65; 95% CI, 0.53–0.77; *p* = 0.021) but improved for non-septic AKI survivors (AUC, 0.83; 95% CI, 0.70–0.97; *p* = 0.003)
Wilson et al. [22]	Plasma	Randox’s multiplexed Biochip Arrays	500 AKI	11	sTNFR1; sTNFR2; CYSTATIN C; NGAL	A multivariable panel containing sTNFR1, sTNFR2, cystatin C, and eGFR discriminated between those with and without kidney disease progression (AUC 0.79 [95% CI, 0.70–0.83]). Optimization of the panel showed 95% sensitivity and a negative predictive value of 92% used to stratify patients at low risk for disease severity.
Merchant et al. [26]	Urine	ELISA	15 AKI 32 non-AKI	29	HRG; CFB; CD59; C3; AGT	Two proteins, HRG and CFB were upregulated in AKI patients, showing moderate predictive performance (AUC 0.79; 95% CI, 0.65–0.94; *p* = 0.001 and AUC 0.75; 95% CI, 0.57–0.93; *p* = 0.007). Significant improvement in the risk prediction for primary outcome was observed, specifically for NRI, IDI in addition to CFB and HRG. Only HRG was a significant predictor in the 21 patients with AKI defined by KDIGO criteria.
Coca et al. [27]	Serum	Randox’s multiplexed Biochip Arrays	769 AKI 769 non-AKI	2	sTNFR1; sTNFR2	Plasma sTNFR1 and sTNFR2 measured 3 months after discharge were associated with renal deterioration independent of AKI (HR 4.7, 95% CI, 2.6–8.6) and significant association with renal failure. In this regards, clinical classifier performance was with AUC of 0.83. There was also association of the both biomarkers with Heart failure ((sTNFR1-1.9 (95% CI, 1.4–2.5) and sTNFR2-1.5 (95% CI, 1.2–2.0)) and death ((sTNFR1- 3.3 (95% CI, 2.5–4.2) and sTNFR2-1.5 (95% CI, 1.9–3.1)).
Jiang et al. [28]	Urine	LC-MS/MS	90 CP-AKI	12	GDF15; LRG1; SPP1	Urinary proteomic profiles of GDF15 (1.77-fold) and LRG1 (4.25-fold) were significantly elevated by CP treatment compared to the baseline.
Di Leo et al. [29]	Urine/serum	NephroCheck^®^ (NC) Immunoassay	719 patients at ICU	2	TIMP-2; IGFBP7	TIMP-2 and IGFBP7 levels yielded good performance in prediction AKI development at first 4 days at ICU and in all critically ill patients (AUC of 0.65). The Kaplan-Meier analysis predicted lower risk for AKI development only for those patients who NC test was negative.
Navarrete et al. [30]	Urine/serum	ELISA assay	21 AKI 21 non-AKI	1	PLA2G15/LPLA2	Urinary PLA2G15/LPLA2 activity was associated with subsequent AKI development during/ongoing CPB. There was similar association with PLA2G15/LPLA2 activity from serum. No association was observed between PLA2G15/LPLA2 activity from both biofluids, suggesting that this biomarker might be an early sign of renal response to CPB events.
Navarrete et al. [31]	Urine	Nano RPLC-MS/MS	8 AKI 8 non-AKI	28	KRK1	Investigation on KLK1, confirmed the activity of this enzyme in AKI and non- AKI patients. In fact, increased action of KLK1 was confirmed only in AKI patients who arrived at ICU and had highest activity in comparison to other enzymes, hence providing novel finding related to intraoperative events in human ischemia reperfusion injury during CPB.

LC-MS/MS—lliquid chromatography coupled with tandem mass spectrometry; ELISA—enzyme-linked immunosorbent assay; CI-AKI—contrast-induced acute kidney injury; AUC—area under the curve; 95% CI—confidence interval at 95%; CPB—cardiopulmonary bypass; VI-AKI Vancomycin-associated AKI; HC—healthy volunteer; OR—odd ratios; KDIGO—kidney disease: improving global outcomes; CP-AKI—cisplatin-induced acute kidney injury; ICU—intensive care unit; HR—hazard ratio.

## Data Availability

Not applicable.

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
