# Peer review of "Recent Advances of Proteomics in Management of Acute Kidney Injury"

_diagnostics, 2023, doi:10.3390/diagnostics13162648_

Round 1

Reviewer 1 Report

More discussion on the main topics, proteomics and AKI, should replace those on urological diseases and CKD in the abstract and introduction. The content and organization of the abstract and introduction could be improved.

Line 48: "filtrated"  or "unfiltered"?

Use of pronouns in the manuscript should be corrected (e.g., his in line 122).

I don't see a reason for differentiating sections 3.1 and 3.2.

Definitions of abbreviations (a list or footnote) used in table 1 would be helpful.

The biomarkers in section 3.2 should be organized logically (study type, area of kidney affected, cell types involved, relevance, biomarker type, etc.) or match the presentation in table 1 to facilitate comprehension. Dividing section 3.2 into more than 1 section could help. A diagram that summarizes the biomarkers discussed would be helpful.

Line 137-138 is confusing.

Line 257: concentrations or concertations?

Line 273: Reference 91 might be the wrong reference.

Lines 421-436: The wrong references might have been cited. HRG should be reference 185 according to table 1.

Line 451: proteme?

Some discussion on novel techniques used in biomarker identification would be helpful.

More discussion on the use and challenges of machine learning in biomarker discovery would be helpful.

moderate editing of English language required (especially use of pronouns and spelling).

Author Response

Point-by-point response to reviewers’ comments

We thank the reviewer for the time and for the valuable comments/suggestions. The quality of the manuscript is significantly improved. Please find below the responses to your comments.

Comment 1: More discussion on the main topics, proteomics and AKI, should replace those on urological diseases and CKD in the abstract and introduction. The content and organization of the abstract and introduction could be improved.

Response 1: We thank the reviewer for the contractive suggestion. We agree with his proposal to do structural changes and re-organization of the content in the abstract as well as in the introduction. In the abstract, we remove the part for the urological disease and provided more information about AKI, disease characteristics, molecular mechanism involved, current diagnosis and the role of proteins in disease management. All of this is stated on page 1, line 9-28: “ Acute Kidney Injury (AKI) is currently recognized as a life-threatening disease, leading to an exponential increase in morbidity and mortality worldwide. At present, AKI is characterized by a significant increase in serum creatinine (SCr) levels, typically followed by a sudden drop in glomerulus filtration rate (GFR). Changes in urine output are usually associated with the renal inability to excrete urea and other nitrogenous waste products, causing extracellular volume and electrolyte imbalances. Several molecular mechanisms were proposed to be affiliated with AKI development and progression, ultimately involving renal epithelium tubular cell-cycle arrest, inflammation, mitochondrial dysfunction, the inability to recover and regenerate proximal tubules, and impaired endothelial function…”

As suggested by the reviewer, we have re-constructed the introduction part, providing information on AKI disease  and more information on proteomics and their role in clinical proteomics. Those changes can be seen on page 1, line 33-93: “ In the recent years, there has been a steady and substantial increase of patients suffering from acute kidney injury (AKI), affecting 13.3 million people worldwide with a mortality rate of up to 1.7 million deaths [1, 2].This complex disorder accounts many pathophysiological distinct conditions, and it is still considered as under-recognized outcome, usually associated with secondary aetilogies like cardiovascular complications or sepsis [3]. By definition, AKI is characterized by a significant reduction of the renal function and a subsequent increase in serum creatinine levels (SCr ≥ 26.4 μmol/L), associated with short- and/or long-term complications. Usually, the early signs originate in the proximal tubular cells of the renal cortex, where symptoms are asymptomatic until disease progression is advanced [4]. The spectrum of kidney injuries is manifested within hours or a few days without reduced urine output. The outcome is extremely serious, causing the accumulation of unfiltrated waste blood products, impaired electrolyte homeostasis, and inflammation, which in turn, induce an imbalance of normal kidney function [5]…..”

We believe that now we have addressed the comments accordingly. Thank you.

Comment 2: Line 48: "filtrated" or "unfiltered"?

Response 2: This was a typo and now has been corrected.

Comment 3: Use of pronouns in the manuscript should be corrected (e.g., his in line 122).

Response 3:  This is correct, and we thank the reviewer for the comment. We have now corrected all typos, pronouns, wording ect. In addition, the manuscript has been read and edited by a native English speaker.

Comment 4: I don't see a reason for differentiating sections 3.1 and 3.2.

Response 4: Thank you for this suggestion. We agree with the reviewer, and we have corrected it.

Comment 5: Definitions of abbreviations (a list or footnote) used in table 1 would be helpful.

Response 5: The reviewer is correct with his statement that abbreviations will be helpful and more understandable for the readers. We have added in a footnote under the table 2.

Comment 6: The biomarkers in section 3.2 should be organized logically (study type, area of kidney affected, cell types involved, relevance, biomarker type, etc.) or match the presentation in table 1 to facilitate comprehension. Dividing section 3.2 into more than 1 section could help. A diagram that summarizes the biomarkers discussed would be helpful.

Response 6: Re-organization and restructuring of the manuscript was crucial. We really appreciate this valuable suggestion. For improving the quality of the manuscript, we have generated a new table in which the biomarkers were grouped by type, study, affected area of the kidney, affected kidney cell types and AKI category. This is presented on page 4, line 127 under section 3.1. Additionally, we have generated a new figure where percentage of biomarker distribution in the AKI type is shown on page 4, line 133. Furthermore, the main part/section was re-organized and discussed based on the biomarker type: diagnostic, prognostic and monitoring. For this part, separate titles and paragraphs were generated (starting from page 11, line 150 until page 19 line 503). Now, we believe that we have addressed the reviewer comments and improved the quality of the manuscript. Thank you.

Comment 7: Line 137-138 is confusing.

Response7: This statement and the part for NGAL has been shorten. Also, we have moved that paragraph in the last section. On page 21, line 562-531 we stated: “One example is Neutrophil gelatinase-associate lipocalin (NGAL) also known as siderocalin or lipocalin 2 (LCN 2) who is an iron-carrying protein with molecular mass of 25 kDa. NGAL can be detected in urine and plasma [204]. Several studies confirmed the NGAL ability of proactive action in immune response to bacteria, prevention of apoptosis and proliferation of renal tubular cells. In fact, this represents one of the possible molecular pathways for preservation and defense of the kidney normal homeostasis during AKI onset, but it is non-specific biomarker related to other renal and non-renal conditions [205, 206].”

Comment 8: Line 257: concentrations or concertations?

Response 8: This was a typo. Now is corrected. Thanks.

Comment 9: Line 273: Reference 91 might be the wrong reference.

Response 9: The reference has been checked and it is correctly referenced.

Comment 10: Lines 421-436: The wrong references might have been cited. HRG should be reference 185 according to table 1.

Response10: That is true, and it has been corrected.

Comment 11: Line 451: proteme?

Response 11: This was a typo. Now is corrected. Thanks.

Comment 12: Some discussion on novel techniques used in biomarker identification would be helpful.

Response 12: We thank the reviewer for this positive suggestion. The discussion on current and novel proteomic techniques has been provided. A paragraph was generated on page 20, line 532-556: ‘’ Regardless of their application, mass spectrometry (MS) has developed continuously, hence providing the opportunity for faster throughput for top-down and bottom–up MS-based methods, deeper proteome coverage and higher sensitivity in measurements of thousands of proteins in thousands of samples. This is achieved by combining different mass analyzers, Quadrupole Oribitrap with the asymmetric track lossless (Astral) analyzer, providing >200 Hz MS/MS scanning speed, high sensitivity, and a narrow mass accuracy window [197]. Protein pathway array (PPA) is another proteomic platform used for screening and analysis of protein signatures that control cancer development. This is a gel-based technique that is mainly used for tissue proteomics, where antibody mixtures are employed to detect specific antigens in protein samples derived from a biopsy or tissue [198]….”

We hope that now the reviewer suggestion is properly addressed.

Comment 13: More discussion on the use and challenges of machine learning in biomarker discovery would be helpful.

Response 13: This is also very nice suggestion from the reviewer point of view. For the reason to improve the quality of the manuscript and to be more close to the general population, we have included additional paragraph where we discussed the challenges of machine learning application for biomarker discovery and validation. We believe the most important points on ML in proteomics studies have been considered and discussed. This paragraph is in the last section, on page 21, line 575-596: ” Simultaneously, the generation of high-resolution proteomic data has largely increased the computational power and clearly demonstrated the utility of ML algorithm deployment in various applications, especially in the identification of protein signatures as well as the prediction of disease course based on proteome and clinical data. On the other hand, extracting meaningful biological information from the large dataset using ML is challenging. Most importantly, common pitfalls include imprecise study design with uncertain primary or secondary outcomes and data that are biased and not harmonized across different patient cohorts during biomarker discovery and validation studies….”

Thank you very much for the constructive comment.

Comment 14: Comments on the Quality of English Language moderate editing of English language required (especially use of pronouns and spelling).

Response: We have to admit that the reviewer was right. Due to the close timeline for submission of the manuscript, we did not have opportunity to improve the language. However, the manuscript in the revision process was read and edited by a native English speaker. Thank you.

Reviewer 2 Report

The paper is very good, but I recommend the authors add more block diagrams and flow charts to enrich their paper and make it more readable for readers>

Check the fonts on all papragraph

Add conclusion  part .

Author Response

Point-by-point response to reviewers’ comments

We thank the reviewer for the time and for the valuable comments/suggestions. The quality of the manuscript is significantly improved. Please find below the responses to your comments.

Comment 1: 1The paper is very good, but I recommend the authors add more block diagrams and flow charts to enrich their paper and make it more readable for readers

Response: We thank the reviewer for the valuable suggestion. We agree with it and for improvement of the manuscript quality, one additional table and 2 figures were generated. Also, the main part was re-organized by grouping the biomarkers based on the function: diagnostic, prognostic and monitoring. Thank you very much.

Comment 2: Check the fonts on all paragraph

Response 2: This was checked and corrected.

Comment 3: Add conclusion part

Response3: Thank you for this suggestion. We have generated the conclusion part and further provide summary and future outlook where we also discussed the current novel proteomic platforms and challenges of machine learning approaches in biomarker discovery and validation studies.  

Round 2

Reviewer 1 Report

The authors significantly improved the organization and discussion of the manuscript. They also included additional sections and figures that improve the quality of the manuscript. However, typos and grammar issues (e.g., assigning genders to cellular components) are still significantly present in the manuscript, although they do not always prevent comprehension of the involved sentences. Also, there are 2 Table 1 in the manuscript, and it would be better for the paragraph in lines136-142 to precede the first Table 1 (The list of biomarkers evaluated in AKI clinical studies). It is unclear what "shows the biomarker characteristic" (line 127, Table 1 footnote) means. Organizationally, it would be better for Figure 3 to appear closer to the last paragraph (lines 597-598).

Minor editing of English language required.

Author Response

Point-by-point response to the reviewer’s comments

Once again, we would like to thank the reviewer for the thoughtful comments and for the efforts to improve the quality of our manuscript. Please find below the response to the raised concern.  

Comment 1: The authors significantly improved the organization and discussion of the manuscript. They also included additional sections and figures that improve the quality of the manuscript. However, typos and grammar issues (e.g., assigning genders to cellular components) are still significantly present in the manuscript, although they do not always prevent comprehension of the involved sentences. Also, there are 2 Table 1 in the manuscript, and it would be better for the paragraph in lines136-142 to precede the first Table 1 (The list of biomarkers evaluated in AKI clinical studies). It is unclear what "shows the biomarker characteristic" (line 127, Table 1 footnote) means. Organizationally, it would be better for Figure 3 to appear closer to the last paragraph (lines 597-598).

Response 1: We apologize for the inconsistency present in our manuscript. The reviewer is correct. There were multiple typos and technical mistakes as a result of working with several versions of the manuscript. Now, we have carefully inspected, corrected, and verified the whole text in the manuscript. Also, biomarkers in Table 1 are sorted based on their role: diagnostic, prognostic and monitoring. We hope that this concern has been properly addressed. Thank you for the constructive suggestion.
